# β-Caryophyllene: A Single Volatile Component of *n*-Hexane Extract of *Dracaena cinnabari* Resin

**DOI:** 10.3390/molecules25214939

**Published:** 2020-10-26

**Authors:** Mohamed Al-Fatimi

**Affiliations:** Department of Pharmacognosy, Faculty of Pharmacy, Aden University, P.O. Box 5411 (Maalla), Aden 00967-2, Yemen; alfatimi@web.de

**Keywords:** *Dracaena cinnabari*, resin, β-caryophyllene, GC-MS, Soqotra

## Abstract

The pure Soqotri resin of *Dracaena cinnabari* Balf.f. (Dracaenaceae) has no volatile smell due to its low content of volatile constituents. Although it is insoluble in *n*-Hexane, we found that the resin, when suspended in *n*-Hexane within five days at 5 °C, led to the extraction of a small portion of a single volatile liquid constituent, which was identified by GC-MS as sesquiterpene β-caryophyllene. This method of extracting the volatile constituents using hexane under cooling is very suitable for resins of the *Dracaena* species because these resins usually contain very few volatile terpenes and/or non-terpenes, and they may contain only one volatile terpene per resin as this study result. β-Caryophyllene was identified and separated for the first time from the Soqotri standard resin of *Dracaena cinnabari*. Therefore, β-caryophyllene, as a new chemical property, can support to evaluate the purity of the Soqotri resin. Moreover, a big mass of *D. cinnabari* resin can yield concentrated β-caryophyllene as a liquid extract for further pharmaceutical and nutraceutical applications.

## 1. Introduction

Several phenolic derivatives have been isolated and identified in the resins of different species of *Dracaena*; for example, *D. cinnabari* resin, which mostly contains flavonoids [1]. The volatile content of the resin has been rarely studied due to the difficulty of the solubility of the resin in nonpolar solvents such *n*-Hexane and due to the very low content of the volatile oil in the resin that cannot be obtained by the method of water distillation. However, the headspace method has been previously used to identify some volatile terpenes in the resin [2]. In this study, we studied a method to obtain the volatile content of the *D. cinnabari* resin by *n*-Hexane extraction for five days under low temperature.

β-Caryophyllene has been reported as an antimicrobial, antioxidant, and cytotoxic agent [3,4,5]. These results explain that the combination of β-caryophyllene and polyphenols can insert a strong synergist effect on the biological and pharmacological activities of the Soqotri resin including its activities against cancer cell lines [6].

## 2. Results and Discussion

The Soqotri standard resin of *Dracaena cinnabari* is insoluble in *n*-Hexane [1] due to its rare content of volatiles. This was identified by the non-volatile odor of the pure resin [1]. However, in the present study, we found that by suspending the powdered resin in the *n*-Hexane at a low temperature of 5 °C for five days, it could release a pure compound that dissolved in a small portion in the *n*-Hexane. This *n*-Hexane filtrate of the resin was investigated by GC-MS. The result identified β-caryophyllene as a single volatile hydrocarbon component that formed the liquid part (100% volume) of the volatile content. The calculated kovats retention index (RI) was 1416 compared to the (RI) reference of 1417, according to Adam, 2007 [7]. β-Caryophyllene (trans-(1R,9S)-8-methylene-4,11,11-trimethylbicyclo [7.2.0] undec-4-ene) was identified as a yellowish volatile liquid hydrocarbon bicyclic sesquiterpene (Figure 1, Table 1). The mass spectrum of the identified β-caryophyllene peak was compared to the reference mass spectrum of β-caryophyllene found in the NIST (National Institute of Standards and Technology) Mass Spectral Library (Figure 2).

The investigation of a resin filtrate of a non-polar solvent such as *n*-Hexane under cooling might contain liquid volatile constituents, even if the resin is insoluble in this solvent. It can separate a volatile pure compound or mixture of volatile constituents. These volatile constituents can be identified in the solvent filtrate by GC-MS.

By the resin suspension in *n*-Hexane for five days at 5 °C, it reduced the volatile oxygenated terpenes that are still solid by cooling. In contrast to the volatile oxygenated terpenes, the hydrocarbon terpenes remained in liquid state even with cooling. Therefore, the liquid hydrocarbon terpenes can be dissolved in the *n*-Hexane, and separated from the resin corpus.

In this study, β-caryophyllene is reported for the first time in the resin of *D. cinnabari.* On the other hand, this volatile sesquiterpene could not be identified in the resin of the *D. cinnabari* using the method of headspaces HS-SPME/GC, but other volatile monoterpenes have been identified [2]. This difference may be attributed to the use of different types of resins and different extraction methods. According to our previous study, there are many types of Soqotri resin obtained from *D. cinnabari* where the pure resin has no characteristic odor or only very weak volatile odor [1]. This explains the identification of only a single hydrocarbon sesquiterpene in the Soqotri pure resin, which we used in this study.

On the other hand, caryophyllene oxide, as a derivative of β-caryophyllene, was identified in the resin of other *Dracaena* species: *D. draco* subsp. *ajgal* from Morocco using headspaces HS-SPME/GC [2]. Nevertheless, the use of *n*-Hexane in the resin of *D. cochinchinensis* from China led to the identification of the existence of only three different volatile sesquitepenes: τ-cadinol, τ-muurolon, and α-cadinol [8]. This confirmed the results that the pure resins of different *Dracaena* species often contain volatile sesquiterpenes. This result can be used as analytical evaluation for the resin of *D. cinnabari*. Moreover, β-caryophyllene was previously identified in the fruit and leaf of *Dracaena draco* using the HS-SPME/GC method [9,10] and in the leaf of *Dracaena reflexa* using the distillation method [11] (Table 2).

## 3. Conclusions

In general, resins of *Dracaena* species contain very few volatile terpenes with little content. Therefore, these resins are either slightly soluble or insoluble in some lipophilic solvents such as *Dracaena cinnabari* resin in *n*-Hexane (1). The suspension of powdered resin in *n*-Hexane for some days under cooling led to the extraction of the liquid hydrocarbon sisquiterpenes from the resin. In our study, we identified β-caryophyllene by GC-MS as a single sesquiterpene in the resin of *D. cinnabari*.

β-Caryophyllene, as a new chemical constituent, can assist in evaluating the purity of the Soqotri resin of *D. cinnabari*. β-Caryophyllene can be separated from the resin of *D. cinnabari* by *n*-Hexane and concentrated for further pharmaceutical applications.

## 4. Materials and Methods

### 4.1. Plant Material

*D. cinnabari* resin was collected from Soqotra Island, southern Yemen. The plant was identified by the Pharmacognosy Department, Aden University, Yemen. A voucher specimen (No. MF-SOC 001) has been deposited at the personal herbarium of the author.

### 4.2. Extraction

Ten grams of the dried and powdered resin was suspended in the 50 mL *n*-Hexane for five days at 5 °C and filtrated. The filtrate without reducing was kept at 4 °C until analysis by GC-MS.

### 4.3. GC-MS Analysis

The *n*-Hexane filtrate of *D. cinnabari* was analyzed by an analytical GC-MS system consisting of an Agilent 6890N gas chromatograph and a mass selective detector (Agilent^®^5973 Network MSD, Agilent Technologies, Santa Clara, CA, USA). Injection was done with an Agilent^®^7683 Series Injector (Split 1:40 at 250 °C, 2.0 µL; carrier gas: helium 1.1 mL/min (60 kPa) at 110 °C; pressure rise: 6 kPa/min). The MS was operated in the electron impact mode with an ionization energy of 70 eV. The oven program started with 1 min at 70 °C, then the oven temperature was increased at 3 °C/min to 220 °C. Full scan mass spectra were acquired from 35–350 *m*/*z* at a rate of 4.5 scans/s and with a 5.00 min solvent delay. Chromatography was performed using a 30 m DB-5 column (J&W Scientific, Folsom, CA, USA) with 0.25 mm i.d. and 0.25 µm film thickness. The detected compounds were identified by processing the raw GC-MS data with ChemStation G1701CA software and comparing them with the NIST mass spectral database 2.0 d (National Institute of Standards and Technology, Gaithersburg, MD, USA) and from the retention indices and mass spectra of standard compounds. Relative amounts of detected compounds were calculated based on the peak areas of the total ion chromatograms (TIC).

## Figures and Tables

**Figure 1 molecules-25-04939-f001:**
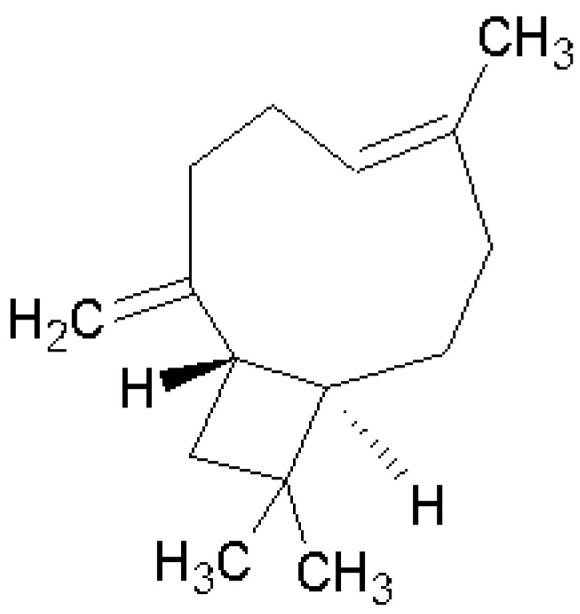
β-Caryophyllene structure.

**Figure 2 molecules-25-04939-f002:**
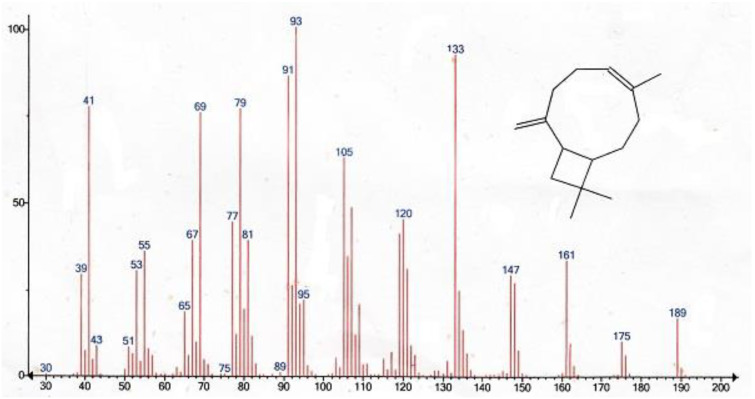
Mass spectrum of the identified β-caryophyllene peak compared to the reference mass spectrum of β-caryophyllene found in the NIST Mass Spectral Library.

**Table 1 molecules-25-04939-t001:** Chemical composition of the *n*-Hexane extract of the resin of *Dracaena cinnabari.*

Retention Time (min)	Compound	Content%	Retention Index RI	Retention Index Reference RI-ref
20.403	β-Caryophyllene	100	1416	1417

RI: Retention index experimentally determined on the DB-5 column relative to C10–C20 n-alkanes, the compound listed in order of its elution on the DB-5 column. RI-ref: Retention index reference according to Adam, 2007 [7].

**Table 2 molecules-25-04939-t002:** Identification of β-caryophyllene in *Dracaena* species using different methods.

*Dracaena* Species	Plant Part/Exudates	Method	Identification of β-Caryophyllene	Reference
*Dracaena cinnabari*	resin	HS-SPME/GC	none	[2]
*D. serrulata*				
*D. ombet*				
*D. draco* subsp. *draco*				
*D. draco* subsp. *ajgal*				
*D. cochinchinensis*	resin	*n*-Hexane extraction	none	[8]
*D. draco*	fruit	HS-SPME/GC	β-caryophyllene	[9]
*D. draco*	leaf	HS-SPME/GC	β-caryophyllene	[10]
*D. reflexa* var. *angustqo*	leaf	steam distillate was extracted into hexane	β-caryophyllene	[11]

HS-SPME/GC: head space, solid phase microextraction with gas chromatography.

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
