# Peer review of "β-Caryophyllene: A Single Volatile Component of n-Hexane Extract of Dracaena cinnabari Resin"

_molecules, 2020, doi:10.3390/molecules25214939_

Round 1

Reviewer 1 Report

In this manuscript, the author presents the identification of b-caryophyllene as the only component of the n-hexane extract from the Dracaena cinnabari resin.

The manuscript is well written (apart from some errors), however there is no special interest in the data presented.

The terpene is identified by GC/MS, is not isolated nor the amount extracted is given.

The author claims that this result could be used as a chemotaxonomic marker, but the same compound was observed in other species of Dracaena although in fruits or leaf, and it can be expected that it could be isolated from the resin if studied using the same methodology. The claim that it could be a source of b-caryophyllene for the industry has no base unless the amount isolated is large, but the author does not give that information, and it seems to be a very small amount.

The results presented in this manuscript by themselves do not constitute enough material for a publication. This data should be part of a larger study.

Author Response

Reviewer 1

1. In this manuscript, the author presents the identification of b-caryophyllene as the only component of the n-hexane extract from the Dracaena cinnabari resin. The manuscript is well written (apart from some errors), however there is no special interest in the data presented.

Answer: errors were corrected

2. The terpene is identified by GC/MS, is not isolated nor the amount extracted is given.

Answer: it was already written that the compound is identified and not isolated.

3. The author claims that this result could be used as a chemotaxonomic marker, but the same compound was observed in other species of Dracaena although in fruits or leaf, and it can be expected that it could be isolated from the resin if studied using the same methodology.

Answer: The comparison was between the resin types of Dracaena species and NOT between all parts of the Dracaena species including the resins. The parts of a plant such as leaf and fruit are organized parts but the resin is unorganized part; it is a product of a plant get by e.g. incision of Dracaena cinnabari bark.

 However, according your comment, I revised the phrase as followings: In the abstract:  “Therefore, β-caryophyllene was found to be as a new chemical property to evaluate the Soqotri resin”. In the conclusions: “β-Caryophyllene as a new chemical constituent was found to evaluate the pure Soqoti resin of D. cinnabari”.

4. The claim that it could be a source of b-caryophyllene for the industry has no base unless the amount isolated is large, but the author does not give that information, and it seems to be a very small amount.

Answer: Because the Soqotri resin is found in the local markets in big mass. However, we write the word “applications” instead technology and added “….as liquid extract”.

5. The results presented in this manuscript by themselves do not constitute enough material for a publication. This data should be part of a larger study.

Answer: This work is prepared as a “short note” or “short communication”; to give new data and suggested method for further investigations of volatile constituents of different resins not only  from species of Dracaena but also from other plant species in different world localities.

Reviewer 2 Report

The conducted study allowed for the identification and separation of β-Caryophyllene from Soqotri standard resin of Dracaena cinnabari.

It is a contribution that broadens the current state of knowledge.

The study was performed and described in an appropriate manner.

The manuscript is also properly formatted.

I propose to accept the work in the present form to Molecules journal.

Author Response

It is a contribution that broadens the current state of knowledge.

The study was performed and described in an appropriate manner.

The manuscript is also properly formatted.

I propose to accept the work in the present form to Molecules journal.

Answer: best thanks for the reviewer

Reviewer 3 Report

The Dracaena cinnabari plant lives on the island of Socotra (between Yemen and the Horn of Africa) and has been exploited for millennia for its essences. In fact, when the bark or leaves are cut, they secrete a resin used in ancient times for many purposes: as a dye, in medicine, and as incense. However, this resin, due to its low content of volatile constituents, is difficult to characterize. In this brief communication, for the first time, the volatile components were extracted from the resin using hexane under cooling. The subsequent characterization by GC-MS allowed to identify the aromatic terpene beta-caryophyllene. In the literature, various biological activities are attributed to beta-caryophyllene, such as anti-inflammatory, antibiotic, antioxidant, anti-cancer and local anesthetic activities.

The experimental strategy adopted in this short communication is interesting for the extraction from matrices containing few volatile constituents. Furthermore, the beta-caryophyllene identified in Soqotri resin of Dracaena cinnabari could be used as a chemotaxonomic marker to differentiate the types of resins, but also extracted from the resin for various applications.

The paper presented one brief communication on a new approach to obtain a pure compound from the Soqotri resin of Dracaena cinnabari. This resin is normally insoluble in hexane, due to its low content of volatile substances. However, the author was able to isolate and identify beta-caryophyllene using low-temperature hexane.
I believe that the data may be of interest to operators in the sector. Furthermore, since it is a brief communication and a single experimental approach, I think that the paper can be added the GC-MS chromatogram of the beta-caryophyllene identified

Author Response

Reviewer 3

The Dracaena cinnabari plant lives on the island of Socotra (between Yemen and the Horn of Africa) and has been exploited for millennia for its essences. In fact, when the bark or leaves are cut, they secrete a resin used in ancient times for many purposes: as a dye, in medicine, and as incense. However, this resin, due to its low content of volatile constituents, is difficult to characterize. In this brief communication, for the first time, the volatile components were extracted from the resin using hexane under cooling. The subsequent characterization by GC-MS allowed to identify the aromatic terpene beta-caryophyllene. In the literature, various biological activities are attributed to beta-caryophyllene, such as anti-inflammatory, antibiotic, antioxidant, anti-cancer and local anesthetic activities.

The experimental strategy adopted in this short communication is interesting for the extraction from matrices containing few volatile constituents. Furthermore, the beta-caryophyllene identified in Soqotri resin of Dracaena cinnabari could be used as a chemotaxonomic marker to differentiate the types of resins, but also extracted from the resin for various applications.

The paper presented one brief communication on a new approach to obtain a pure compound from the Soqotri resin of Dracaena cinnabari. This resin is normally insoluble in hexane, due to its low content of volatile substances. However, the author was able to isolate and identify beta-caryophyllene using low-temperature hexane.
I believe that the data may be of interest to operators in the sector. Furthermore, since it is a brief communication and a single experimental approach, I think that the paper can be added the GC-MS chromatogram of the beta-caryophyllene identified.

Answer: best thanks for the reviewer.

Reviewer 4 Report

Manuscript ID: molecules-937326

After careful review of the ms entitled “β-Caryophyllene a single volatile component of n-hexane extract of Dracaena cinnabari resin” by Al-Fatimi (Manuscript ID: molecules-937326), this reviewer thinks that the topic of the study is very interesting, but several points of this manuscript should be revised.

The Author shows that the volatile composition of the n-hexan extract from D. cinnabari resin is represented by β-Caryophyllene and proposes this sesquiterpene for species identification of Dracaena species. In this regards, the critical point of this study is the lack of a comparison among different Dracaena species by presence/absence determination of β-Caryophyllene in the n-hexan extracts from resin of other species. Additionally, β-Caryophyllene was recently found by Vaníčková et al. [2] (as cited by this work of Al-Fatimi) in the resin extract of another Dracaena species: D. draco subsp. ajgal from Marocco (DDA) as reported in Table 1 of [2], but not reported in Table 2 of the present work when referred to [2]. On the contrary, β-Caryophyllene was not found by Vaníčková et al. [2] in the resin extract of D. cinnabari from Socotra (DC) as reported in Table 1 of [2]. There is another discordance, RI can be used for terpene identification, but Vaníčková et al. [2] reported RI = 1574 of β-Caryophyllene on DB-5 column, while in this work Al-Fatimi reported RI = 714 of β-Caryophyllene on DB-5 column. Considering all these facts, it is evident that the identification and the proposed role of β-Caryophyllene as chemotaxonomic marker are disputable.

In the present form the ms is not sufficiently clear and readable. This reviewer recommends the authors to revise the ms by improving English language and removing inaccuracies.

Author Response

Reviewer 4

1. After careful review of the ms entitled “β-Caryophyllene a single volatile component of n-hexane extract of Dracaena cinnabari resin” by Al-Fatimi (Manuscript ID: molecules-937326), this reviewer thinks that the topic of the study is very interesting, but several points of this manuscript should be revised.

The Author shows that the volatile composition of the n-hexan extract from D. cinnabari resin is represented by β-Caryophyllene and proposes this sesquiterpene for species identification of Dracaena species. In this regards, the critical point of this study is the lack of a comparison among different Dracaena species by presence/absence determination of β-Caryophyllene in the n-hexan extracts from resin of other species.

 Answer: According your comment and comment from reviewer 1, I revised the phrase as followings:

In the abstract:  “Therefore, β-caryophyllene was found to be as a new chemical property to evaluate the Soqotri resin.”

In the conclusions: “β-Caryophyllene as a new chemical constituent was found to evaluate the pure Soqoti resin of D. cinnabari”.

2. Additionally, β-Caryophyllene was recently found by Vaníčková et al. [2] (as cited by this work of Al-Fatimi) in the resin extract of another Dracaena species: D. draco subsp. ajgal from Marocco (DDA) as reported in Table 1 of [2], but not reported in Table 2 of the present work when referred to [2]. On the contrary, β-Caryophyllene was not found by Vaníčková et al. [2] in the resin extract of D. cinnabari from Socotra (DC) as reported in Table 1 of [2].

Answer: Caryophyllene oxide and NOT β-Caryophyllene, which was found by Vaníčková et al. [2] in resin of other species of Dracaena. Please check the table 2 in this reference. However, we can add according to your comment this phrase: “Caryophyllene oxide as a derivative of β-Caryophyllene was identified in resin of another Dracaena species: D. draco subsp. ajgal from Marocco [2].”

3. There is another discordance, RI can be used for terpene identification, but Vaníčková et al. [2] reported RI = 1574 of β-Caryophyllene on DB-5 column, while in this work Al-Fatimi reported RI = 714 of β-Caryophyllene on DB-5 column. Considering all these facts, it is evident that the identification and the proposed role of β-Caryophyllene as chemotaxonomic marker are disputable.

Answer: Vaníčková et al. identified Caryophyllene oxide: RI=1574 but NOT β-Caryophyllene RI = 714; please check the content of the table 1 in the reference [2].  

4. In the present form the ms is not sufficiently clear and readable. This reviewer recommends the authors to revise the ms by improving English language and removing inaccuracies.

Answer: mistakes were  corrected.

Round 2

Reviewer 1 Report

The author has improved the manuscript by correcting the English and overall presentation.

The claims that were maid about the use of the beta-caryophillene extracted as a biomarker for the characterization of the Soqotri resin and the possible use in the industry have been changed to something that can be sustained in the evidence presented.

Thus, although this manuscript does not contain too many relevant data, can be published as a short note

Author Response

Response to reviewer 1

The author has improved the manuscript by correcting the English and overall presentation. The claims that were maid about the use of the beta-caryophillene extracted as a biomarker for the characterization of the Soqotri resin and the possible use in the industry have been changed to something that can be sustained in the evidence presented. Thus, although this manuscript does not contain too many relevant data, can be published as a short note

Answer: Best thank for the reviewer

Reviewer 4 Report

After a second review of the ms plants-937326, the ms was changed, but English language should be further improved and the concepts should be rewritten more clearly. This reviewer recommends the author to revise the ms by improving English language and removing inaccuracies (L28, L31-32, etc.).

Because the Retention Index (RI) can be used for terpene identification then there is a strong discrepancy with other Authors. Vaníčková et al. [2] reported RI = 1574 of Caryophyllene oxide on DB-5 column, while in this work Al-Fatimi reported RI = 714 of β-Caryophyllene on DB-5 column. In http://dx.doi.org/10.1016/j.bjp.2017.02.004, de Alencar Filhoa et al. (2017) reported RI values similar for Caryophyllene oxide (RI=1586-1582) and β-Caryophyllene (RI=1421-1417) on DB-5; likewise, in Rec. Nat. Prod. (2008) 2:2 33-38, Noudogbess et al. reported similar RI values for Caryophyllene oxide (RI=1577) and β-Caryophyllene (RI=1425) on DB-5; similarly, in DOI: http://dx.doi.org/10.1002/ffj.1221, Zrira et al. (2003) reported RI values for Caryophyllene oxide (IK=1581) and β-Caryophyllene (IK=1419) on DB-5. Consequently, in the present work, Author reported the greatly different value of RI = 714 for β-Caryophyllene on DB-5 column. It is evident that the identification of β-Caryophyllene is questionable, please solve and discuss this issue.

Moreover, Author could add the GC-MS chromatogram of the β-Caryophyllene

Author Response

Response to reviewer 4

After a second review of the ms plants-937326, the ms was changed, but English language should be further improved and the concepts should be rewritten more clearly. This reviewer recommends the author to revise the ms by improving English language and removing inaccuracies (L28, L31-32, etc.). Because the Retention Index (RI) can be used for terpene identification then there is a strong discrepancy with other Authors.  Vaníčková et al. [2] reported RI = 1574 of Caryophyllene oxide on DB-5 column, while in this work Al-Fatimi reported RI = 714 of β-Caryophyllene on DB-5 column.  In http://dx.doi.org/10.1016/j.bjp.2017.02.004, de Alencar Filhoa et al. (2017) reported RI values similar for Caryophyllene oxide (RI=1586-1582) and β-Caryophyllene (RI=1421-1417) on DB-5; likewise, in Rec. Nat. Prod. (2008) 2:2 33-38, Noudogbess et al. reported similar RI values for Caryophyllene oxide (RI=1577) and β-Caryophyllene (RI=1425) on DB-5; similarly, in DOI: http://dx.doi.org/10.1002/ffj.1221, Zrira et al. (2003) reported RI values for Caryophyllene oxide (IK=1581) and β-Caryophyllene (IK=1419) on DB-5.  Consequently, in the present work, Author reported the greatly different value of RI = 714 for β-Caryophyllene on DB-5 column. It is evident that the identification of β-Caryophyllene is questionable, please solve and discuss this issue. Moreover, Author could add the GC-MS chromatogram of the β-Caryophyllene

Answer: Thank you for the pointing out the error. The RI of β-Caryophyllene was already written correct in the text and in table 1 as following “The Kovats-Index was 1416 equaled the references 1467 in www.flavornet.org and www.pherobase.com”. This was revised into “The calculated kovats retention index (RI) was 1416 compared to the (RI) reference 1467 in www.flavornet.org and www.pherobase.com”.  In the table 1 in the first column, the value 714 was found for MF= RMF= 714 but not for RI, therefore, this error is deleted. I added also the retention time for the peak and the GC-MS chromatogram of the β-Caryophyllene..